# MedRAG: Enhancing Retrieval-augmented Generation with Knowledge Graph-Elicited Reasoning for Healthcare Copilot

Submission Id: 2870

## Abstract

Retrieval-augmented generation (RAG) is a well-suited technique for retrieving privacy-sensitive Electronic Health Records (EHR). It can serve as a key module of the healthcare copilot, helping reduce misdiagnosis for healthcare practitioners and patients. However, the diagnostic accuracy and specificity of existing heuristic-based RAG models used in the medical domain are inadequate, particularly for diseases with similar manifestations. This paper proposes MedRAG, a RAG model enhanced by knowledge graph (KG)-elicited reasoning for the medical domain that retrieves diagnosis and treatment recommendations based on manifestations. MedRAG systematically constructs a comprehensive four-tier hierarchical diagnostic KG encompassing critical diagnostic differences of various diseases. These differences are dynamically integrated with similar EHRs retrieved from an EHR database, and reasoned within a large language model. This process enables more accurate and specific decision support, while also proactively providing follow-up questions to enhance personalized medical decision-making. MedRAG is evaluated on both a public dataset DDXPlus and a private chronic pain diagnostic dataset (CPDD) collected from our cooperated hospital, and its performance is compared against various existing RAG methods. Experimental results show that, leveraging the information integration and relational abilities of the KG, our MedRAG provides more specific diagnostic insights and outperforms state-of-the-art models in reducing misdiagnosis rates. Our code will be available at *https://github.com/username00-c/MedRAG.git*

## CCS Concepts

• **Applied computing → Health care information systems**; • **Information systems → Language models**.

## Keywords

Heathcare Copilot, Retrieval-augmented Generation, Knowledge Graph, Large Language Models, Decision Support

**ACM Reference Format:**
Anonymous Author(s), Submission Id: 2870. 2018. MedRAG: Enhancing Retrieval-augmented Generation with Knowledge Graph-Elicited Reasoning for Healthcare Copilot. In *Proceedings of Make sure to enter the correct conference title from your rights confirmation emai (Conference acronym 'XX).* ACM, New York, NY, USA, 15 pages. https://doi.org/XXXXXXX.XXXXXXX

## 1 Introduction

Diagnostic errors cause significant harm to healthcare systems worldwide. In the United States, approximately 795,000 individuals each year suffer permanent disability or death due to misdiagnosis of dangerous diseases. These errors are predominantly attributed to cognitive biases and judgmental mistakes [9, 37]. "Healthcare Copilot" is a medical AI assistant designed to provide diagnostic decision support, mitigating biases and increasing efficiency for healthcare practitioners, while also empowering patients and improving overall decision-making [1, 2, 28, 45, 46]. We conducted interviews to gather requirements and suggestions from users of the healthcare copilot. The results showed that one of the most important and challenging tasks for a healthcare copilot is to provide an accurate diagnosis based on patient manifestations [1], followed by offering appropriate treatment plans and medication recommendations based on the diagnosis. In addition, when patient information is insufficient or the diagnosis is ambiguous, the healthcare copilot should proactively offer precise follow-up questions to enhance the decision-making process [3, 25, 38, 46, 66].

Retrieval-augmented generation (RAG) offers an advanced approach by utilizing domain-specific, private datasets to address user queries without the need for additional model training [12, 17, 29]. This approach is well-suited for retrieving information from privacy-sensitive Electronic Health Records (EHRs), and helps healthcare professionals to reduce the risk of misdiagnosis as a healthcare copilot [22, 59]. The existing medical RAG and LLMs fine-tuned on medical datasets often rely on heuristic-based approaches, leading to incorrect or vague outputs, particularly when diseases share similar manifestations, making differentiation difficult. [18, 24, 31, 58, 63, 67] as shown in Figure 1(a). To address this, we introduce MedRAG, a framework that combines RAG with a comprehensive diagnostic knowledge graph, enabling more accurate reasoning and tailored treatment recommendations by grounding predictions in structured, inferable medical data [21, 26, 34, 54]. This approach significantly enhances the reasoning ability of RAG, enabling it not only to identify subtle diagnostic differences but also to proactively infer relevant follow-up questions, further clarifying ambiguous patient information, as shown in Figure 1(b).

Specifically, a diagnostic knowledge graph (KG) with a four-tier hierarchical structure is constructed systematically through advanced techniques, including disease clustering, hierarchical aggregation and large language model augmentation. The diagnostic differences KG searching module then identifies all critical diagnostic differences KG related to the input patient by performing multi-level manifestations matching within the diagnostic KG. Finally, a KG-augmented RAG module synthesizes the retrieved EHRs and

---

[1]"Manifestations" typically include all observable signs and symptoms of a patient's condition, such as physical indicators (e.g., rash, fever), patient-reported symptoms (e.g., pain, dizziness), and measurable clinical data (e.g., blood pressure, lab results).

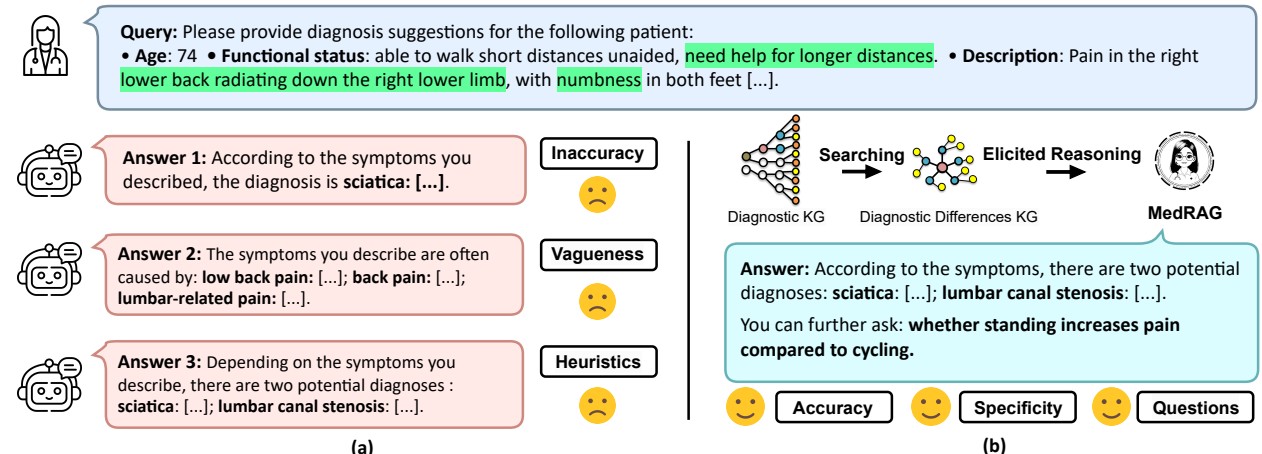

Figure 1: (a) The existing RAG and LLMs rely on heuristic-based approaches, leading to incorrect or vague outputs, particularly when diseases share similar manifestations (high lighted by green colour). (b) MedRAG is a RAG framework with KG-elicited reasoning ability that can make accurate diagnostic decisions and generate highly specific diagnoses, along with proactively providing follow-up questions when necessary.

the critical diagnostic differences KG to elicit the reasoning within a large language model. This integration enhances the system's ability to make precise and highly specific diagnostic decisions, while also providing personalized treatment recommendations, medication guidance, and, when necessary, proactive follow-up questions.

We evaluate the general applicability of MedRAG by a public dataset DDXPlus [14] and real-world clinical applicability by a private chronic pain diagnostic dataset (CPDD). Performance is quantitatively compared against several popular state-of-the-art (SOTA) RAG models, including FL-RAG [44] and DRAGIN [50]. We further validate the generalization of MedRAG on widely-used open-source LLMs, including Mixtral-8x7B[20] and Llama-3.1-Instruct[11], as well as on some closed-source LLMs such as GPT-3.5-turbo[39], GPT-4o[40]. Experimental results demonstrate that our model outperforms existing RAG approaches in terms of diagnostic accuracy and specificity. Additionally, MedRAG demonstrates robust generalization across various LLMs, and proves highly effective in generating reasoning-based follow-up diagnostic questions. These capabilities are particularly valuable for distinguishing between diseases with similar manifestations. Based on extensive experiments, our key contributions can be summarized as follows:

- We deliver two diagnostic knowledge graphs: one focused on chronic pain and the other based on DDXPlus [14], a large-scale synthesized dataset. These knowledge graphs contain a rich hierarchical structure of diseases, along with their key diagnostic differences. This comprehensive organization allows for enhanced precision in disease differentiation and diagnosis, enabling better decision-making support across various medical systems.
- We proposed a novel RAG approach enhanced by KG-elicited reasoning, which significantly improves RAG's ability to make accurate and highly specific diagnostic decisions. In addition to supporting personalized treatment recommendations and medication guidance, it proactively generates

follow-up questions when necessary. These enhancements greatly optimize the decision-making process in complex medical scenarios.
- Comprehensive experiments conducted on two datasets demonstrate the superiority of our model over existing RAG and LLM approaches. Additionally, the results highlight its applicability across various backbone LLMs and its effectiveness in proactively generating reasoning-based diagnostic questions.

## 2   Related Works

### 2.1   LLMs and RAG in Healthcare

Large Language Models (LLMs) have been increasingly applied to healthcare tasks such as EHR analysis, clinical note generation, virtual medical assistant, and clinical decision support [18, 22, 56, 63, 69]. While LLMs fine-tuned on medical datasets can handle large amounts of unstructured clinical information. However, most of these models are heuristic-based, with limitations such as generating incorrect or vague information and struggling to handle complex patient cases [24, 63]. To address this, integrating external information sources becomes essential to improve their contextual accuracy. We adopt a Retrieval-Augmented Generation (RAG) approach [29]. RAG enhances LLMs by incorporating retrieved text passages from external sources such as electronic health records, medical papers, textbooks, and databases into their input, resulting in significant improvements in knowledge-intensive tasks [4]. In the field of healthcare, integrating retrieved information grounds the predictions in current, verifiable medical data, resulting in more accurate, specificity and context-aware outputs such as diagnostic assessments and treatment recommendations. RAG typically employs a retrieve-and-read approach to retrieve information based on the initial user query and an answer is generated using that content [13, 16, 27, 48, 49, 70]. However, this simplicity restricts

their ability to adapt to complex and evolving medical cases. Enhanced RAG models aim to improve retrieval and generation quality by integrating more sophisticated components such as retrievers, re-rankers, filters, and readers [7, 23, 29, 35, 47, 64]. Despite these advancements, delivering accurate clinical decision support remains challenging. The models often struggle to provide precise diagnoses, particularly when diseases share similar manifestations, making differentiation difficult. Our proposed MedRAG addresses these challenges by systematically constructing a four-tier hierarchical diagnostic knowledge graph to elicit reasoning for the generation module of RAG. This approach enables the model to make accurate diagnostic decisions and generate highly specific diagnoses along with personalized treatment recommendations.

## 2.2 Knowledge Graph-enhanced LLMs and RAG

Recent studies have focused on creating strategies that integrate knowledge graphs to enhance LLMs and RAG, enabling them to generate accurate and reliable medical responses. Compared to knowledge contained in document repositories [19], knowledge graphs offer structured and inferable information, making them more suitable for augmenting LLMs and RAG [21, 26, 30, 34, 54, 65, 72]. Several works [21, 33, 53, 60, 68, 73] propose training sequence-to-sequence models from scratch, focusing on dialogue generation by conditioning the output on entities extracted from knowledge graphs. However, existing medical knowledge graphs [5, 15, 54, 71] often fall short because they lack the detailed and structured information necessary for accurate diagnostic assistance, especially when distinguishing between diseases with similar manifestations. To overcome this limitation, we introduce MedRAG, a framework that combines RAG with a comprehensive diagnostic knowledge graph to enhance the reasoning ability of RAG in identifying subtle differences in diagnoses. MedRAG allows physicians to input patients' medical records or manifestations. Our knowledge graph is constructed based on patterns extracted from Electronic Health Record (EHR) databases and augmented by LLMs, making it highly scalable and adaptable to various medical specialties. It supports customization with local databases, ensuring relevance to specific clinical settings. We employ LLMs to enrich the knowledge graph by providing detailed descriptions of the manifestations of each disease at the leaf nodes, including symptoms, affected areas, activity limitations, and other pertinent features.

## 3 Preliminaries

**Definition 3.1 (Diagnostic Knowledge Graph).** Given an EHR database $D$ and an LLM $\mathcal{M}_a$, our target is to construct a four-tier hierarchical diagnostic knowledge graph $\mathcal{G}$. A multi-hop path, from the top level to the bottom level of $\mathcal{G}$ is represented as ($E_{L1} \xleftarrow{r_s} E_{L2} \xleftarrow{r_s} E_{L3} \xrightarrow{r_m} E_{L4}$). $E_{L3}$ is the set of all diseases (i.e. potential diagnoses) names extracted from $D$, $E_{L2}$ represents the set of subcategories of $E_{L3}$, and $E_{L1}$ is the set of broader categories of $E_{L2}$. Each $e_{Lij}$ is a disease name or a category name and $e_{Lij} \in E_{Li}$. $E_{L1}$ and $E_{L2}$ are generated by hierarchical aggregation in Section 4.1.1, they indicate the diseases with similar manifestations. $r_s$ is an "is_a" relation, indicating a hierarchical or subordinate relationship. $r_m$ is a "has_manifestation_of" relation between diseases

and their manifestations. $E_{L4}$ contains two subtypes: $E_{L4a}$, representing disease-specific features augmented by the LLM $\mathcal{M}_a$, and $EL4d$, representing features decomposed from the manifestations extracted from the EHR database $D$.

**Definition 3.2 (Diagnostic Differences KG Searching).** Given a $\mathcal{G}$ and the input patient's manifestations $q$, let $e_{L2_s} \in E_{L2}$ denote a certain subcategory identified through the method described in Section 4.2.3 determined from $q$. The target is to extract the diagnostic differences KG $K$, related to $e_{L2_s}$, from $\mathcal{G}$.

**Definition 3.3 (RAG).** We define a typical retrieval-augmented generation approach for generating diagnostic reports in two phases: algorithm $\mathcal{R}$ for the retrieval phase and LLM $\mathcal{M}_g$ for the generative phase. A prompt $p_{naive}$ is used to guide $\mathcal{M}_g$ to generate the final report. Given a $q$, $D$ and embedding model $\mathcal{E}$, $\mathcal{R}$ retrieves top-$k$ relevant documents $d_r$, and then $\mathcal{M}_g$ generates answer $A$ with $q$, $d_r$ and prompt $p_{naive}$ as shown in Equation 1 and 2:

$$d_r = \mathcal{R}(q, D, \mathcal{E}), \tag{1}$$

$$A = \mathcal{M}_g(q, d_r, p_{naive}). \tag{2}$$

## 4 Methods

In this section, we elaborate on the details of our proposed MedRAG, and the overall framework is illustrated in Figure 2. MedRAG includes five modules:

- **Input**: The input to MedRAG is the description of patient manifestations, which can be either structured EHR or unstructured text descriptions.
- **Output**: The output of MedRAG includes the diagnoses, treatment recommendations, medication guidance and follow-up questions when necessary.
- **Diagnostic Knowledge Graph Construction**: This module constructs a four-tier hierarchical diagnostic knowledge graph systematically. First, potential diagnoses and corresponding manifestations are extracted from an EHR database to form a four-tier disease KG through clustering and hierarchical aggregation. Then, an LLM is used to augment the graph with critical diagnostic differences, transforming it into a diagnostic KG.
- **Diagnostic Differences KG Searching**: This module identifies key diagnostic differences by decomposing patient manifestations into clinical features, such as symptoms and locations, through medical chunking. Then, the extracted features are embedded and matched with relevant diagnostic differences via multi-level matching and upward traversal within the diagnostic KG.
- **KG-elicited Reasoning RAG**: This module comprises a document retriever and a KG-elicited reasoning LLM engine. The retriever selects top-k relevant EHRs based on patient embeddings and integrates them with critical diagnostic differences KG to trigger reasoning in the LLM, generating final diagnoses and providing medical recommendations.

Next, we will provide a detailed explanation of each phase in the following subsections.

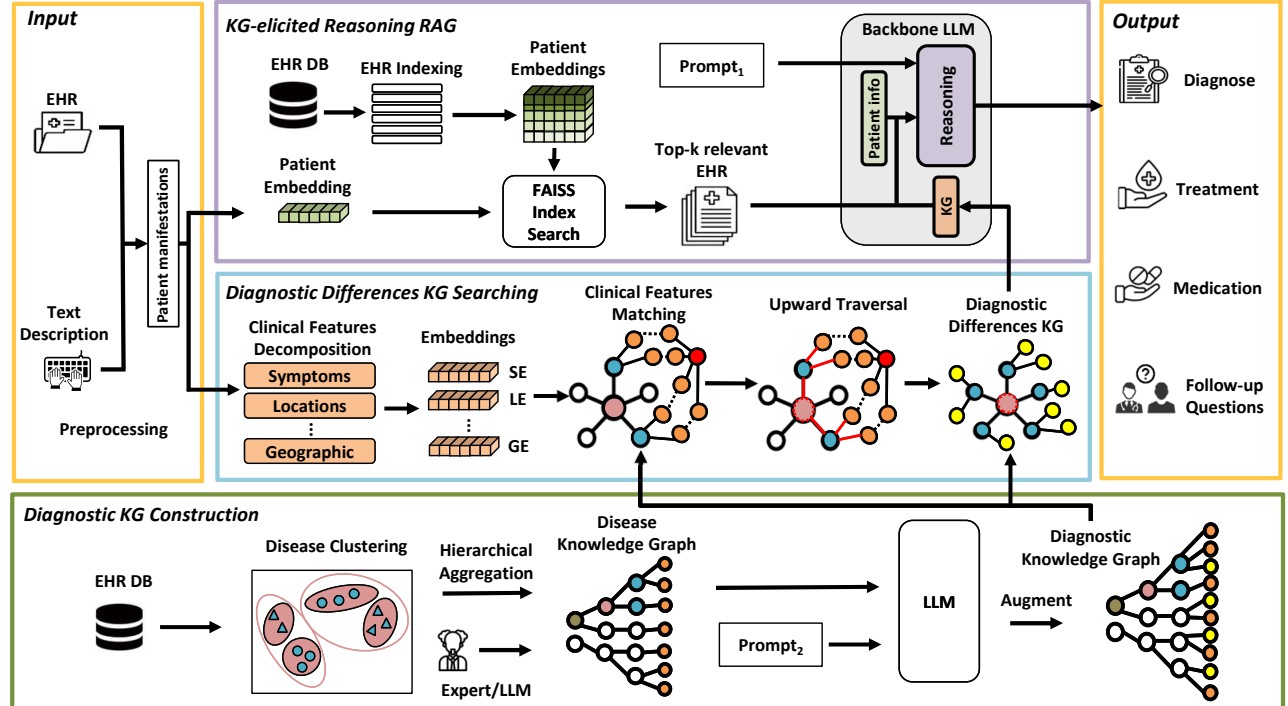

Figure 2: The overall framework of MedRAG. MedRAG first extracts patient (red node) manifestations from structured or unstructured input, and decomposes different clinical features. These features are embedded and matched with a diagnostic KG to identify critical diagnostic differences KG. MedRAG's KG-elicited reasoning RAG module retrieves relevant EHRs and integrates them with these diagnostic differences KG to trigger reasoning in an LLM. This reasoning generates precise diagnoses, treatment recommendations, and follow-up questions.

## 4.1 Diagnostic Knowledge Graph Construction

To enhance the reasoning capabilities and fill the knowledge gaps of the RAG, we propose constructing a diagnostic knowledge graph $\mathcal{G}$ tailored to the medical domain of a specific EHR database. The construction of the diagnostic knowledge graph draws inspiration from the hierarchical structure of the World Health Organization's International Classification of Diseases, 11th Edition (ICD-11) [43] [2].

*4.1.1 Disease Knowledge Graph Construction.* The forms and representations of the diseases in an EHR database are diverse, we first unify the set of original disease descriptions $E_{L3_{raw}}$ by disease clustering to $E_{L3}$. The most common disease name within each cluster is regarded as the final disease name and is assigns to all other diseases in the cluster, as shown in Equation 3:

$$E_{L3} = C(E_{L3_{raw}}, \mathcal{E}), \tag{3}$$

where $C$ represents the clustering model applied to $E_{L3}$, $\mathcal{E}$ is an embedding model.

Then we use the unified $E_{L3}$ to construct a four-tier hierarchical disease knowledge graph through hierarchical aggregation. This

graph integrates the relationships between diseases and their potential categories, with each disease aggregated into a subcategory and category. We define the disease knowledge graph as $\mathcal{G}_D$ where $\mathcal{G}_D \subset \mathcal{G}$ aggregated by $\Theta$ and LLM $\mathcal{M}_h$, as shown in Equation 4:

$$\mathcal{G}_D = \Theta(E_{Li}, \mathcal{M}_h, \mathcal{E}), i = 3, 2. \tag{4}$$

In the first phase, we apply LLM-based topic aggregation using $\mathcal{M}_h$, which extracts the most relevant topics from $EL3$ to aggregate subcategories. These subcategory topics are then further aggregated into higher-level categories, forming the hierarchical structure from subcategories to broader categories. Next, hierarchical clustering is applied to assign diseases in $E_{L3}$ into aggregated subcategory topics and then subtopics to topics.

This approach leverages LLM's powerful semantic understanding and topic extraction capability, allowing for a more nuanced categorization of diseases in topic aggregation. By applying hierarchical clustering to the LLM-based topics, diseases in $E_{L3}$ are aggregated into a hierarchical structure. Hierarchical aggregation introduces multiple layers of granularity to $E_{L3}$, ensuring that diseases with different manifestations are properly categorized.

To effectively utilize historical diagnoses from $D$ as accurate representations of disease manifestations, we decompose their manifestations of the diseases in $E_{L3}$, parsing them into discrete features $E_{L4d}$. Every single feature like symptom, location, or activity limitation from each $e_{L3_i}$ is created as a node $e_{L4d_i} \in E_{L4}$. This final

---

[2] The specific classification principles of our diagnostic KG and ICD-11 are different. Our approach classifies and organizes diseases based on the similarity of their manifestations, rather than ICD-11's traditional classification based on diagnostic categories. As a result, while the hierarchical concept is similar, the ICD-11 structure cannot be directly applied to our model.

decomposition results in the comprehensive disease knowledge graph $\mathcal{G}_D$, capturing both disease category information derived from hierarchical aggregation and their associated features.

*4.1.2 Knowledge Graph Manifestation Augmentation.* The knowledge in $\mathcal{G}_D$ only contains information from $D$, which is insufficient to accurately diagnose all diseases, particularly when distinguishing between diseases with similar clinical manifestations. Therefore, the integration of external knowledge is essential. To complement the diagnostic knowledge graph with essential knowledge that is not present in $D$, we augment external knowledge $E_{L4}$ to $\mathcal{G}_D$ that aids in distinguishing diseases with similar manifestations. We traverse all disease $e_{L3_i}$ and employ a prompt $p_a$ specially tailored for searching and generating the nuances of the diseases on an LLM denoted by $\mathcal{M}_a$. As shown in Equation 5, each generated diagnostic key difference node $e_{L4a_{ij}}$ is then connected to its corresponding $e_{L3_i}$ with relationship $r_m$. Thus we obtain a chain $E_{L3} \xrightarrow{r_m} E_{L4a}$. For example, we generate a manifestation and relation to disease node *lumbar spondylosis* and form a chain: $<$ *lumbar_spondylosis, has_symptom, stiffness_or_pain_in_the_l−ower_back* $>$.

$$\{e_{L3_i}\}_{i=1}^n \xrightarrow{\mathcal{M}_a(p_a, e_{L3_i})} \{e_{L4a_{ij}}\}_{i=1, j=1}^{n, m_i}, \quad (5)$$

$$E_{L4} = E_{L4d} \cup E_{L4a}, \quad (6)$$

$$\mathcal{G} = \mathcal{G}_D \cup_{E_{L3}} \{E_{L3} \cup E_{L4}\}_{i=1}^n, \quad (7)$$

where $\mathcal{M}_a$ and $p_a$ represent the large language model for disease manifestation augmentation and its prompt respectively.

The finalized four-tier hierarchical diagnostic knowledge graph $\mathcal{G}$ is formed by integrating the disease knowledge graph $\mathcal{G}_D$ with $E_{L4}$ combined with $E_{L4_a}$ and $E_{L4_d}$, as shown in Equation 6 and 7.

## 4.2 Diagnostic Differences KG Searching

*4.2.1 Decomposition of Manifestations.* Given $q$ as a query, which is a description of the patient's manifestations, we perform sentence trunking on $q$ to decompose the manifestation into more detailed features, denoted as $f_1, f_2, \ldots, f_n \in q$. We define a mapping function to describe the process, shown in Equation 8:

$$q \xrightarrow{\phi} \{f_1, f_2, \ldots, f_n\}. \quad (8)$$

*4.2.2 Clinical Features Matching.* Given a $q$, we compute the semantic similarity score $sim$ between $f_i$ and $e_{L4d_i}$, shown in Equation 9:

$$sim_{ij} = \mathcal{S}(f_i, e_{L4d_j}, \mathcal{E}), \quad (9)$$

where $\mathcal{S}$ is similarity model and $\mathcal{E}$ is embedding model applied to $f_i$ and $e_{L4d_j}$ before similarity calculation.

For each patient feature $f_i$, we retrieve the top-$m$ most similar $e_{L4d_j}$, where $m$ denotes the number of closest matches selected. Totally, the system retrieves $n \times m$ matching nodes in the $\mathcal{G}$. To address the scenario where a $f_i$ has no closely matching counterpart in $E_{L4d}$, we introduce an indicator function $\delta(sim_{ij}, t_{matching})$ to filter irrelevant matches:

$$\delta(sim_{ij}, t_{matching}) = \begin{cases} sim_{ij} & \text{if } sim_{ij} > t_{matching} \\ 0 & \text{otherwise,} \end{cases} \quad (10)$$

$$T = \bigcup_{i=1}^n \left\{ e_{L4d_j} \mid j \in \arg \max_{j' \in \{1, \ldots, |E_{L4d}|\}}^m \delta(sim_{ij}, t_{matching}) \right\} \quad (11)$$

where $T$ represents the set of nodes $e_{L4d_j}$ that satisfy the condition $sim_{ij} > t_{matching}$. The indicator function $\delta$ ensures that only $e_{L4d_j}$ with a similarity score above the threshold are selected into $T$. Through clinical features matching, we successfully matched $q$ to the most relevant clinical feature nodes in $\mathcal{G}$.

*4.2.3 Upward Traversal.* To precisely match the patient's most relevant $e_{L2_s}$, we employ upward traversal which determines the closest disease subcategory by aggregating votes based on the shortest path distances between $t_i \in T$ and $e_{L2_j}$ in the graph.

For $t_i$, we calculate the shortest path to each disease subcategory $e_{L2_j}$ by upward traversing through the graph. Denote the shortest path distance between $t_i$ to $e_{L2_j}$ as $P(t_i, e_{L2_j})$. If $e_{L2_{ik}}$ represents the closest disease subcategory node for the current $t_i$, the vote count for $e_{L2_{ik}}$ is incremented by one. We then accumulate the votes for each $e_{L2_{ik}}$ during the reversal and identify the node with the highest vote count as the $e_{L2_s}$. This voting process is formalized through the indicator function $\chi$, defined as follows:

$$\chi(t_i, e_{L2_{ik}}) = \begin{cases} 1 & \text{if } e_{L2_{ik}} = \arg \min_{e_{L2_j}} P(t_i, e_{L2_j}), \\ 0 & \text{otherwise} \end{cases} \quad (12)$$

$$e_{L2_s} = \arg \max_{e_{L2_{ik}}} \sum_{t_i \in T} \chi(t_i, e_{L2_{ik}}), \quad (13)$$

Taking this $e_{L2_s}$ as the parent node, we traverse downward towards $E_{L4}$, retrieving all $e_{L3_i}$ that are adjacent to $e_{L2_s}$ and their adjacent $e_{L4a_i}$. Given $e_{L2_s}$, let $E_{L3_s} = \{e_{L3_i} \mid e_{L3_i} \in \text{Adj}(e_{L2_s})\}$ denote the set of disease nodes that belong to $e_{L2_s}$. Similarly, define $E_{L4a_s} = \{e_{L4a_i} \mid e_{L4a_i} \in \text{Adj}(e_{L3_i}), e_{L3_i} \in E_{L3_s}\}$ to denote the set of feature nodes linked to the disease nodes in $E_{L3_s}$.

We concatenate all triples $(e_{L3_s}, r_m, e_{L4a_i})$, where $e_{L3_s} \in \text{Adj}(e_{L2_s})$ and $e_{L4a_i} \in \text{Adj}(e_{L3_s})$, to form the set of diagnostic differences KG:

$$K(e_{L2_s}) = \bigcup_{e_{L3_s} \in \text{Adj}(e_{L2_s})} \{(e_{L3_s}, r_m, e_{L4a_s})\}, \quad (14)$$

where $K$ represents the diagnostic differences KG used for the reasoning in the LLM next.

*4.2.4 Proactive Diagnostic Questioning Mechanism.* Inaccurate diagnoses often stem from insufficient or incomplete patient descriptions. To address this issue, we propose a Proactive Diagnostic Questioning Mechanism. When the initial input $q$ lacks some crucial information required for doctors or LLMs to make more precise diagnostic decisions, this mechanism acts as a copilot to cast targeted follow-up questions.

In the diagnostic knowledge graph $\mathcal{G}$, a feature $e_{L4d_i}$ may be connected to multiple disease nodes $e_{L3}$, with each $e_{L4d_i}$ varying in its discriminability. For instance, certain features are more prevalent, such as "pain located in the lumbar region", while others represent more distinctive characteristics, like "pain worsens while walking". Here we define the discriminability score of $e_{L4d_i}$ as the reciprocal of the degree centrality in $\mathcal{G}$:

$$\sigma(e_{L4d_i}) = \frac{n-1}{deg(e_{L4d_i})}, \quad (15)$$

where $n$ represents the total number of $e_{L4d_i}\mathcal{G}$.

We calculate the discriminability score $\sigma(e_{L4d_j})$ for each feature node $e_{L4d_j} \in E_{L4d_s}$ and select those with the highest discriminability scores as follows:

$$\{e_{L4d_{s_1}}, e_{L4d_{s_2}}, \ldots, e_{L4d_{s_k}}\} = \arg \max_{\{e_{L4d_j} | e_{L4d_j} \in E_{L4d_s}\}} \sigma(e_{L4d_j}), \quad (16)$$

where $\{e_{L4d_{s_1}}, e_{L4d_{s_2}}, \ldots, e_{L4d_{s_k}}\}$ represents the selected features with the highest discriminability scores, which are used to proactively guide follow-up questions for clarifying the diagnosis.

### 4.3 KG-elicited Reasoning RAG

KG-elicited Reasoning RAG is the core component of MedRAG, we use an LLM to generate diagnoses, personalized treatment plans, and medication suggestions. Additionally, the system proactively suggests follow-up questions for doctors to clarify missing or ambiguous patient information. As shown in Equation 17, MedRAG utilizes diagnostic differences KG augmented by LLM and a tailored prompt $p_s$ to elicit the reasoning capabilities of LLM.

$$A = \mathcal{M}_g(q, d_r, K, p_s) \quad (17)$$

Unlike most RAG systems that focus on answering short factual questions, our system is tailored for complex tasks in clinical scenarios. The prompts are designed explicitly to optimize the reasoning capabilities of the LLM, particularly in distinguishing between diseases with similar manifestations. The system conducts thorough reasoning by using both the retrieved documents and the diagnostic differences KG extracted from $G$.

We use the EHR database as a document repository to retrieve the most relevant documents $d_r$ corresponding to the patient's manifestations $q$. We then perform a similarity search over the database to identify the most relevant k records. For this, we employ Facebook AI Similarity Search (FAISS) [10], a library optimized for efficient approximate nearest neighbor searches. FAISS allows rapid retrieval of similar records in large-scale EHR datasets, enabling adjustable trade-offs between speed and search accuracy, depending on the size and complexity of the dataset:

After obtaining all inputs, we designed a tailored prompt $p_s$ for guiding the LLM to reason through $K$, generating answers to assist doctors in distinguishing between similar diseases and proactively generating follow-up questions.

## 5 Experiments

### 5.1 Datasets

We evaluate our proposed MedRAG framework using two distinct datasets: one public and one private. The public dataset demonstrates the model's general applicability, while the private dataset, focused on chronic pain patients, enables a more thorough evaluation of MedRAG's diagnostic capabilities in real-world clinical settings.

The public dataset, DDXPlus [14], is a large-scale, synthesized EHR dataset widely recognized for offering complex, diverse medical diagnosis cases. It includes comprehensive patient data such as socio-demographic information, underlying diseases, symptoms, and antecedents. Many studies have employed DDXPlus to benchmark models in medical reasoning and diagnosis [6, 32, 51, 61]. DDXPlus contains 49 different diagnoses with over 1.3 million

patients, each of whom has approximately 10 symptoms and 3 antecedents on average. We ultimately utilized a maximum balanced sub-dataset comprising 13,230 patients' EHRs.

The private dataset is the Chronic Pain Diagnostic Dataset (CPDD), a specialized EHR dataset focused on chronic pain patients. This dataset is collected from a collaborating hospital, it comprises 551 patients with 33 distinct diagnoses. CPDD offers manifestations-specific chronic pain patient data, making it an invaluable resource for testing MedRAG's diagnostic capabilities in clinical settings. For more details on the partitioning, preprocessing, and selection of these two datasets, please refer to the Appendix.

### 5.2 Baselines

In order to explore the performance of the MedRAG, we compare the MedRAG results against six other models:

- **Naive RAG + COT**[57] We apply the chain-of-thought (COT) prompting with a naive RAG model, which only retrieves documents without additional enhancements.
- **FL-RAG**[44] FL-RAG is a multi-round retrieval method that triggers the retrieval module every n tokens.
- **FS-RAG**[52] FS-RAG is an interleaving retrieval method that improves multi-round question answering by alternating between COT reasoning and document retrieval.
- **FLARE**[24] FLARE is an active RAG method that improves knowledge-intensive tasks by retrieving relevant documents when the model encounters uncertain tokens.
- **DRAGIN**[50] DRAGIN is a dynamic retrieval method that enhances language models by retrieving relevant documents based on real-time information needs during generation, triggered by token uncertainty.
- **SR-RAG**[55] In SR-RAG, relevant passages are retrieved from an external corpus based on the initial query and then incorporated into the input of the language model

### 5.3 Experimental Setup

Throughout the experiment, we use text-embedding-3-large [42] from OpenAI as the embedding model $\mathcal{E}$, use IndexFlatIP method after standardization in FAISS index search which is equivalent to cosine similarity in retriever $\mathcal{R}$. In diagnostic differences KG searching, we use K-means algorithm [36] for clustering model $C$ and GPT-4o as LLM $\mathcal{M}_a$ for external diagnostic difference augmentation.

For the quantitative performance comparison, we use Llama-3.1-Instruct 8B [11] as the generative model $\mathcal{M}_g$, as some RAG algorithms only support open-source LLMs. For the generalizability evaluation, we utilize both open-source LLMs, such as Llama-3.1-Instruct 70B [11], Mixtral-8x7B [20], and Qwen2 [62], as well as closed-source models, including GPT-3.5-turbo [39], GPT-4o-mini [41], and GPT-4o [40], as the backbone LLMs. For the hyperparameters of MedRAG, we use $k = 3$ in Retriever and $m = 5$ in diagnostic differences KG searching, set threshold $t$ and $t_{matching}$ to 0.5.

We mask certain existing $e_{L4_s}$ of a patient to simulate scenarios where they are missing. MedRAG then makes diagnoses and generates follow-up questions based on the remaining information. If MedRAG identifies the removed manifestations during questioning, the masked manifestations are added back to the patient's record

as the answer of the patient, and diagnostic reasoning is repeated to evaluate the improvement in diagnostic accuracy. More detail is shown in the Appendix.

## 6 Experimental Results

In this section, we present the results of the experiments to answer the following research questions:

- **RQ1 (Section 6.1)**: Does MedRAG outperform the SOTA RAG methods using the same datasets?
- **RQ2 (Section 6.2)**: Does MedRAG demonstrate compatibility, generalizability and adaptability across different backbone LLMs?
- **RQ3 (Section 6.3)**: Does MedRAG's proactive diagnostic questioning mechanism provide users with impactful, relevant follow-up questions to enhance diagnostic performance?
- **RQ4 (Section 6.4)**: Is the MedRAG system we designed effective, and what is the impact of each module on its overall performance?

### 6.1 Quantitative Comparison (RQ1)

Our experiments evaluate MedRAG against six different SOTA RAG models on 2 two datasets. We report the results using the following metrics: 1) **Accuracy**, defined as the number of correct diagnoses out of the total diagnoses; and 2) **Specificity**, which uses $L1$, $L2$, and $L3$ to represent different diagnostic granularity levels. As outlined in Section 3 (Definition 3.1), $L_i$ refers to the MedRAG select potential diagnoses from $E_{Li}$. This metric evaluates the model's specificity and its ability to differentiate between similar diseases across varying levels of diagnostic granularity.

The result is shown in Table 1, Our proposed MedRAG achieved the best or second-best (with only one exception) performance across multiple metrics in all datasets. Accuracy on the $L3$ metric is the best indicator of MedRAG's performance, as higher specificity increases diagnostic difficulty. MedRAG outperformed the second-best scores on the CPDD and DDXPlus datasets by 11.32% and 1.23%, respectively.

Additionally, most RAG models designed for simpler QA tasks do not perform as well in the more complex medical domain, leading to longer contextual and prompt. These models are often optimized for generating short and straightforward answers, which limits their effectiveness in handling intricate medical queries. We observe models that have a simpler mechanism in the query-organizing phase perform better than part of more sophisticated ones. Except for our MedRAG, models like SR-RAG and FL-RAG also secured several second-best performances. Even the Chain-of-Thought model, which lacks improvements in the retriever or generator components, outperformed some of the other SOTA models in complex medical tasks.

### 6.2 Compatibility, Generalizability and Adaptability (RQ2)

The results in Table 2 demonstrate the performance of incorporating KG-elicited reasoning to various backbone LLMs, including both open-source and closed-source models. The results demonstrate that the inclusion of KG-elicited reasoning significantly enhances

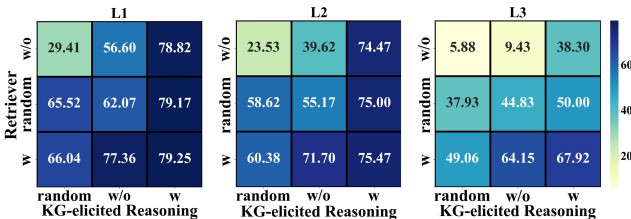

**Figure 3: Ablation result on Llama-3.1-Instruct 8B backbone using the CPDD dataset**

diagnostic accuracy across $L1$, $L2$, and $L3$ for all backbone LLMs, compared to models without its use. For example, Mixtral-8x7B shows a significant $L3$ improvement from 22.34% to 63.46%, demonstrating the effectiveness of our proposed KG-elicited reasoning, particularly in smaller models.

Comparing open-source and closed-source models, the RAG with the GPT-4o as the backbone LLM outperforms all others, showing its superior adaptability with knowledge graph integration. In addition, MedRAG performs best on closed-source models, showcasing our framework's compatibility, generalizability and adaptability. In contrast, token-level RAG models like DRAGIN and FLARE face challenges in adapting to closed-source models due to their inherent frameworks, limiting their potential to achieve better performance across various LLMs.

### 6.3 Proactive Diagnostic Questioning (RQ3)

The results in Table 3 show the impact of following MedRAG's optimized instructive questions and obtaining corresponding patient responses on diagnostic accuracy.

As more detailed information is gathered through these targeted questions, the $L3$ accuracy progressively improves. Initially, with no specific patient information obtained through this questioning process, the $L3$ accuracy is 52.83%, representing MedRAG making a diagnosis with other information with very few manifestations. As the doctor collects more critical details about disease representation, covering from 33.3% to 100% of the key manifestations, the $L3$ score rises from 55.10% to 66.04% and other levels' metrics follow the same trend. This demonstrates the significant effectiveness of MedRAG's proactive diagnostic questioning mechanism, validating its capability to provide doctors with impactful questions that not only enhance diagnostic performance but also improve the efficiency of the medical consultation process.

### 6.4 Ablation Study (RQ4)

We perform ablation studies to evaluate the effectiveness of different components in MedRAG and present the result in Figure 3. Specifically, we assess the retriever component $\mathcal{R}$ and KG-elicited reasoning module $\mathcal{G}$ under three configurations: "random", "with" and "without". In the "random" setting for the retriever $\mathcal{R}$, we choose documents from the entire EHR database randomly. The "without" of the retriever refers to the scenario where no documents are passed to $\mathcal{M}_g$ for diagnostic report generation. The "with" setting of the retriever means to pass the top-$k$ relevant documents to the $\mathcal{M}_g$.

| Method | Model | CPDD | | | DDXPlus | | |
|--------|-------|------|------|------|---------|------|------|
| | | L1 | L2 | L3 | L1 | L2 | L3 |
| Baselines | Naive RAG + COT | 75.47 | 54.72 | 43.40 | 79.28 | 71.89 | 56.84 |
| | FS-RAG | 64.71 | 49.02 | 45.10 | 78.18 | 68.20 | 51.40 |
| | FLARE | 54.84 | 48.39 | 45.16 | 71.09 | 56.70 | 31.02 |
| | FL-RAG | 65.45 | 50.91 | 49.09 | **90.12** | 83.32 | 66.78 |
| | DRAGIN | 78.72 | 59.57 | 40.42 | 80.51 | 70.83 | 50.24 |
| | SR-RAG | 73.58 | 60.38 | 54.72 | 78.65 | 70.28 | 52.16 |
| Ours | MedRAG | **79.25** | **75.47** | **66.04** | 88.65 | **83.46** | **68.01** |

Table 1: Results of quantitative performance comparison

| | Backbone LLMs | Size | w/o KG-elicited Reasoning | | | w/ KG-elicited Reasoning | | |
|--|---------------|------|---------------------------|--|--|--------------------------|--|--|
| | | | L1 | L2 | L3 | L1 | L2 | L3 |
| Open-source Models | Mixtral-8x7B | 13B | 60.38 | 32.08 | 22.34 | 84.62 | 82.69 | 63.46 |
| | Qwen-2.5 | 72B | 66.04 | 41.51 | 39.62 | 80.36 | 73.21 | 64.29 |
| | Llama-3.1-Instruct | 8B | 75.47 | 54.72 | 43.40 | 79.25 | 75.47 | 66.04 |
| | Llama-3.1-Instruct | 70B | 86.79 | 67.92 | 56.60 | 86.79 | **83.02** | 71.70 |
| Closed-source Models | GP-3.5-turbo | - | 83.02 | 56.60 | 45.28 | 70.56 | 68.68 | 50.57 |
| | GPT-4o-mini | - | 88.68 | 67.92 | 56.60 | 85.85 | 75.00 | 60.38 |
| | GPT-4o | - | **90.57** | **71.70** | **60.38** | 91.87 | 81.78 | **73.23** |

Table 2: Performance of MedRAG on different LLM backbones with and without KG-elicited reasoning

| Manifestation Masking Ratio | L1 | L2 | L3 |
|-----------------------------|------|------|------|
| 0% | 60.38 | 56.60 | 52.83 |
| 33.3% | 69.39 | 67.35 | 55.10 |
| 66.6% | 71.43 | 67.35 | 61.22 |
| 100% | 79.25 | 75.47 | 66.04 |

Table 3: Result of proactive diagnostic questioning

For the KG-elicited reasoning module, the "random" configuration denotes randomly selecting subcategory $e_{L2_s}$ and collecting corresponding $K$ accordingly. The "without" is the scenario where no diagnostic differences KG are passed to $\mathcal{M}_g$. Configuration "with" means to pass correct $K$ by the $e_{L2_s}$ to the $\mathcal{M}_g$.

As shown in Figure 3, both the retriever and KG-elicited reasoning module significantly enhance performance across all specificity levels. the best outcomes are achieved when RAG and KG components are combined and aligned, especially for granular diagnosis tasks that demand high specificity.

Notably, randomly selected documents performed better than no documents at all, this phenomenon was explored in detail by [8]. We also observed a performance decline in the lower-granularity levels of $L1$ and $L2$ when transitioning from random to no knowledge from KG when random documents are retrieved. Once correct KG-augmented knowledge was added, this noise effect was mitigated, leading to accuracy improvements across all metrics: an average accuracy increase of 18.88% for $L1$, 26.92% for $L2$, and 18.89% for $L3$, compared to the baseline with random or without KG-elicited reasoning module.

## 7 Conclusion

In conclusion, MedRAG significantly improves diagnostic accuracy and specificity in the medical domain by integrating KG-elicited reasoning with RAG models. By systematically retrieving and reasoning over Electronic Health Records (EHRs) and dynamically incorporating critical diagnostic differences KG, MedRAG offers more precise diagnosis and personalized treatment recommendations. Additionally, MedRAG's proactive diagnostic questioning mechanism proves highly effective, and shows potential capacity to provide doctors and patients with impactful questions that enhance diagnostic performance and improve consultation efficiency. The evaluation of public and private datasets demonstrates that MedRAG outperforms state-of-the-art RAG models, particularly in reducing misdiagnosis rates for diseases with similar manifestations, showcasing its potential as a key module in healthcare copilot. For future work, we aim to further enhance MedRAG's capabilities by incorporating additional multimodal data, such as medical imaging (e.g., MRI), physiological signal data (e.g., ECG), and blood test data to improve diagnostic accuracy and broaden its applicability to a wider range of medical conditions. Additionally, we plan to deploy MedRAG within our healthcare copilot systems (The user interface is shown in the Appendix) for real-world testing in hospitals, ensuring its effectiveness in clinical settings.

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

# Appendix

This appendix is organized as follows:

- **Section A** includes variables and definitions in the paper.
- **Section B** demonstrates the detailed data preprocessing steps and experimental setup, ensuring transparency and reproducibility.
- **Section C** presents intermediate results from experiments.
- **Section D** shows the user interface of healthcare copilot.

## A  Variables and Definitions

The variables used throughout this paper and their definitions are provided in Table A1.

## B  Data Preprocessing and Experimental Setup

### B.1  Settings for Datasets

- **CPDD** We split the data set into a 9:1 ratio for the training set (to be retrieved) and test set. Since the dataset was collected from multiple doctors, the diagnosis descriptions are not standardized. Part of the diagnosis is presented as a type of pain instead of a specific disease. When calculating the accuracy of these pain-type diagnoses, if the predicted result is a disease associated with that type of pain, it will be considered a correct prediction.
- **DDXPlus** We directly use the training set and test set in a split dataset in the ratio of 8:1:1(validation set). Due to the massive size of the dataset with over a million synthesized patients' records, which is too large for the scale of our task, we first fixed the number of samples in the test set to 30, which corresponds to the fewest pathology. For the other pathology with more samples, we randomly select 30 samples to form the whole test set. In the training set, we randomly pick 240 samples for each pathology to retrieve. This approach can ensure we get a maximum balanced sub-dataset containing 13230 patients' EHR in total. The random seed is set to 42.

### B.2  Setup for Proactive Diagnostic Questioning Mechanism

We mask certain existing manifestations of a patient to simulate scenarios where they are missing. MedRAG then generates follow-up questions based on the remaining information. If MedRAG identifies the removed manifestations during questioning, they are added back to the patient's record, and diagnostic reasoning is repeated to evaluate the improvement in diagnostic accuracy.

We begin by selecting all matching manifestation nodes $E_{L4d_s}$ and ranking them according to their discriminability scores. A proportion $r$ of the nodes with the highest discriminability scores is then removed, simulating the scenario where certain key patient features are missing or unclear, shown in Equation 18. After removing, we match the removed nodes $E_{L4d_s}^{\text{del}}$ with each $f_i$, if the similarity score, the corresponding sentence $f_i$ is also removed, as formalized in Equation 19, which simulates the loss of relevant patient information from the input.

$$e_{L4d_s}^{\text{del}} = \text{Top-r}\left(e_{L4d_s}, \sigma(e_{L4d_s})\right),  \quad (18)$$

$$f_i^{\text{del}} = \bigcup_{f_i}\{f_i \mid \mathcal{S}(f_i, e_{L4d_s}^{\text{del}}, \mathcal{E}) > t\}, \quad (19)$$

where $e_{L4d_s}^{\text{del}}$ represents the nodes removed from $e_{L4d_s}$ based on the similarity score threshold $t$ and $f_i^{\text{del}}$ is removed $f_i$.

### B.3  Prompt Engineering

The prompt configuration for the generative model in MedRAG is illustrated in Figure A1. The first block provides instructions as the system prompt. The second block displays the answer template. In the final block, relevant information including the patient's manifestations $q$, retrieved documents $d_r$, and diagnostic differences $K$, is populated in this field.

## C  Intermediate Results

### C.1  Disease Clustering Result

The result of disease clustering in CPDD is Shown in Figure A2. Through the disease clustering operation, we group different forms and representations of the same disease in the EHR database together, assigning a topic to each cluster. This process unifies the representation of diseases, ensuring consistency and comparability. Additionally, it provides a unified foundation for subsequent disease knowledge graph construction and augmentation.

### C.2  Example of Diagnostic Differences Knowledge Graph

While lumbar canal stenosis and sciatica share some similar features, the critical distinguishing factor lies in the response to sitting. In lumbar canal stenosis, features are typically alleviated when sitting, whereas in sciatica, sitting tends to exacerbate the discomfort. The augmented disease features are shown in Figure A3

## D  User Interface (UI)

This section introduces how our MedRAG can be integrated into the user interface design of the healthcare copilot system. The healthcare copilot offers three modes of interaction, as shown in Figure A4.

- **Consultation Mode**: By monitoring the consultation dialogue between the doctor and patient, the system extracts patient manifestations in real-time and provides diagnostic suggestions along with proactive questioning recommendations to guide the consultation.
- **EHR Mode**: By uploading the patient's EHR to the healthcare copilot system, this system automatically extracts the relevant patient manifestations for diagnostic purposes.
- **Typewriting Mode**: The user can manually input the patient's manifestations into the system.

On the results page shown in Figure A5, the output of the healthcare copilot system include diagnoses, instructive follow-up questions, physiotherapy treatments, and medication treatments. This UI integrates the most essential functions derived from extensive interviews we conducted with numerous healthcare practitioners. It ensures that the healthcare copilot system meets the practical needs of healthcare professionals, ultimately enhancing the overall quality of care.

| Variable | Definition |
|---|---|
| $C$ | The clustering model |
| $d_r$ | Retrieved relevant documents |
| $D$ | Electronic Health Record (EHR) database |
| $E_{L1}$ | The set of broad disease categories |
| $E_{L2}$ | The set of disease subcategories |
| $e_{L2_s}$ | The matched subcategory |
| $E_{L3}$ | The set of specific disease names |
| $E_{L3_{raw}}$ | the set of original disease descriptions in $D$ |
| $E_{L3_s}$ | The set of disease nodes connected with $E_{L2_s}$ |
| $E_{L4}$ | The set of disease-specific features |
| $E_{L4_s}$ | The set of features nodes connected with node in $E_{L3_s}$ |
| $E_{L4d_s}^{\text{del}}$ | Deleted features in proactive diagnostic questioning |
| $E_{L4a}$ | Disease-specific features augmented by the LLM |
| $E_{L4d}$ | Features decomposed from the EHR database |
| $e_{L2_{ik}}$ | The closest disease subcategory node |
| $e_{Lij}$ | A disease or category name in the graph where $e_{Lij} \in E_{Li}$ |
| $f_i$ | A specific feature of the patient's manifestation |
| $K$ | The set of diagnostic differences KG identified in the knowledge graph |
| $p_a$ | Prompt used by $\mathcal{M}_a$ for disease manifestation augmentation |
| $p_{naive}$ | Simple prompt used by $\mathcal{M}_g$ |
| $p_s$ | Prompt designed for reasoning and generating diagnostic reports |
| $P$ | The shortest path function |
| $q$ | A input patient's manifestations |
| $r$ | $E_{L4_s}$ removing proportion in proactive diagnostic questioning mechanism |
| $r_m$ | Relation type `has_manifestation_of` between diseases and their manifestations |
| $r_s$ | Relation type `is_a` for hierarchical relationships |
| $sim$ | The similarity score between patient features and nodes in the knowledge graph |
| $T$ | Set of relevant matching nodes in the knowledge graph |
| $t$ | Similarity score threshold in proactive diagnostic questioning mechanism |
| $\chi$ | The voting indicator function |
| $\delta$ | The matching filtering indicator function |
| $\mathcal{E}$ | Embedding model used to compute similarity between features |
| $\mathcal{G}$ | The four-tier hierarchical diagnostic knowledge graph |
| $\mathcal{G}_D$ | The four-tier disease knowledge graph |
| $\mathcal{M}_a$ | LLM used for disease manifestation augmentation |
| $\mathcal{M}_g$ | LLM used for generating diagnostic reports |
| $\mathcal{M}_h$ | LLM used for topic aggregation |
| $\mathcal{S}$ | The similarity model |
| $\phi$ | The decomposition function for $q$ |
| $\sigma(e_{L4d_i})$ | Discriminability score of a feature in the knowledge graph |
| $\Theta$ | The hierarchical aggregation operator |

**Table A1: List of variables and their definitions**

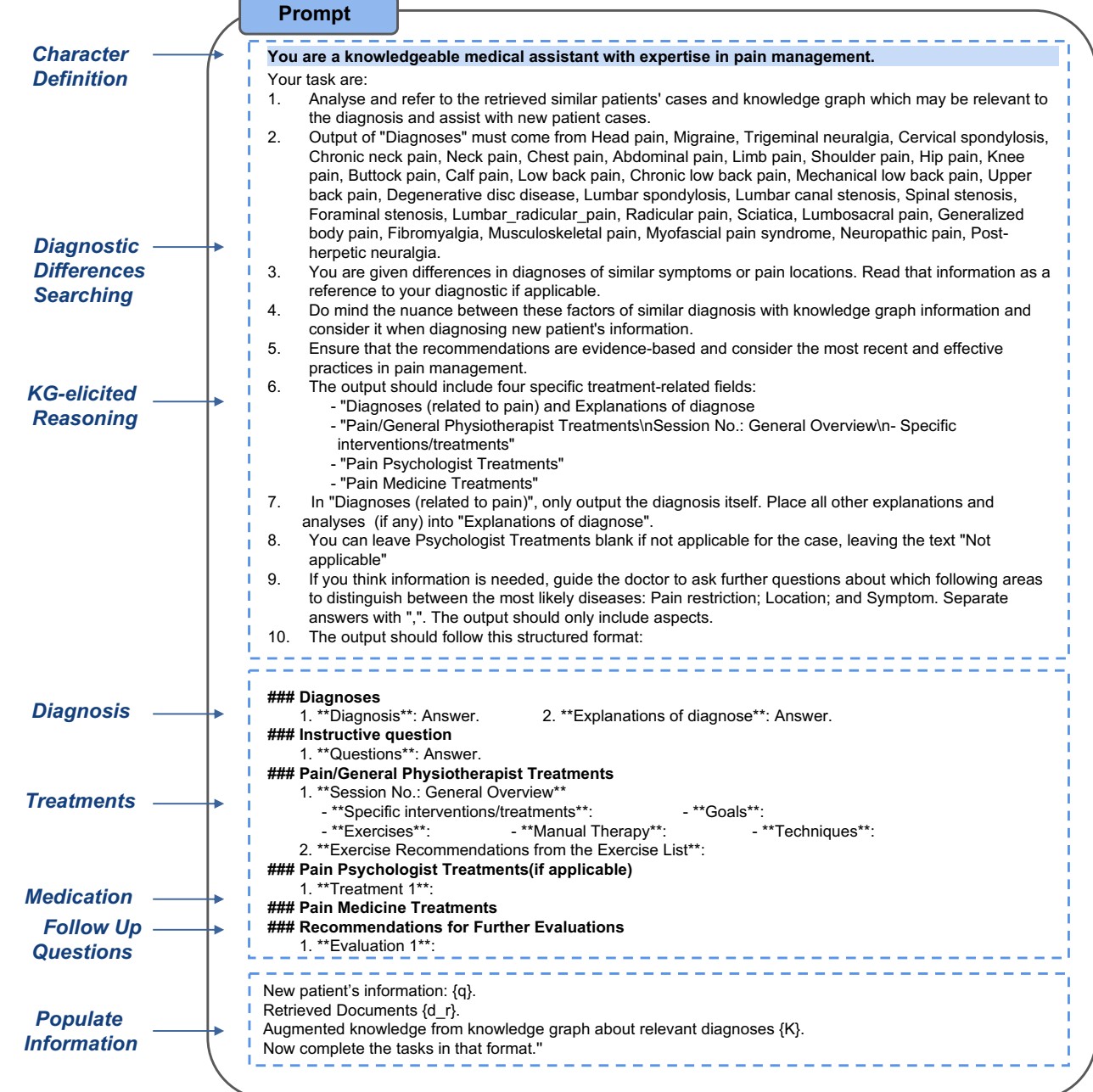

**Figure A1: Prompt for the generative model of MedRAG**

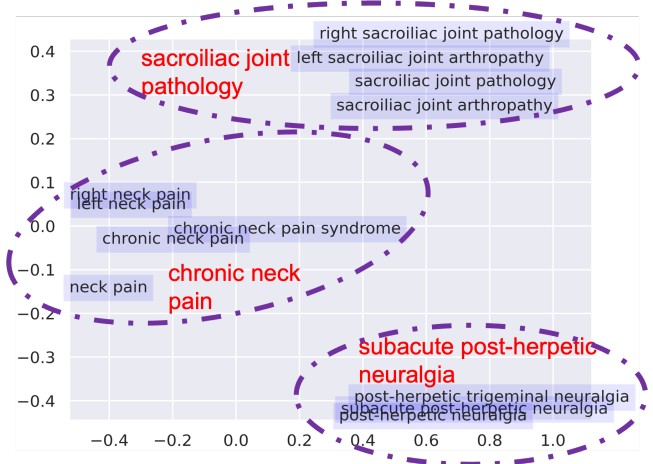

**Figure A2: The result of disease clustering in CPDD**

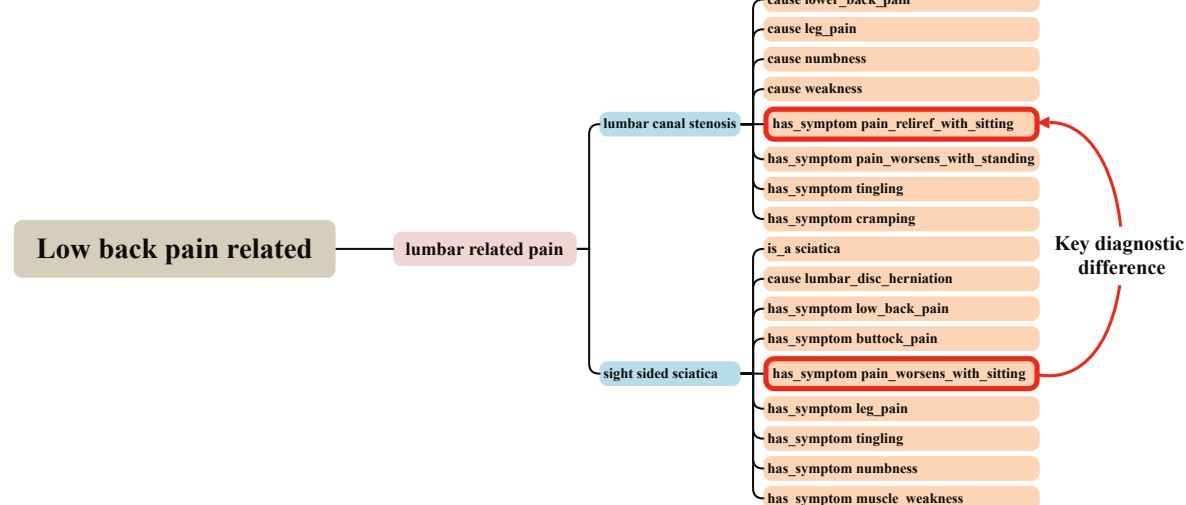

**Figure A3: Diagnostic differences knowledge graph between lumbar canal stenosis and sciatica. (Similar manifestations but opposite responses to sitting (Alleviation vs. Exacerbation))**

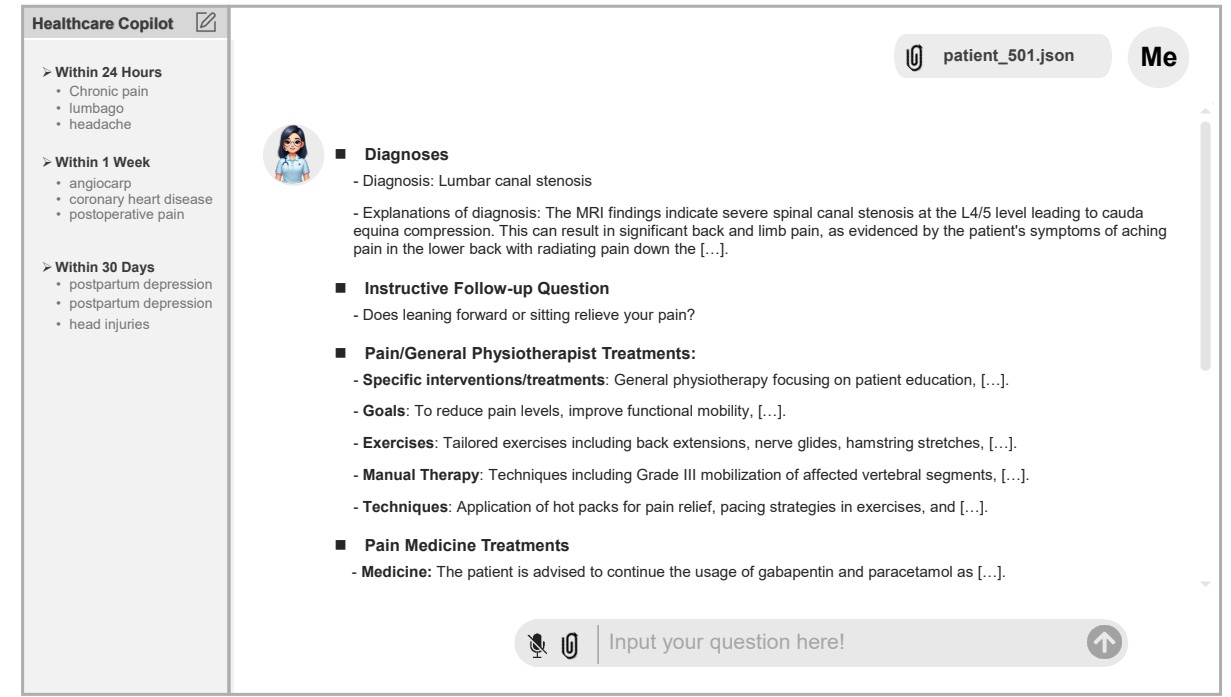

**Figure A4: The interactive interface of healthcare copilot allows multi-turn medical Q&A through voice, files, and text.**

**Figure A5: A specific example of how healthcare copilot could handle the diagnosis of lumbar canal stenosis using a JSON format medical record input, and output relevant treatments.**

