# OpenReview forum: "MedRAG: Enhancing Retrieval-augmented Generation with Knowledge Graph-Elicited Reasoning for Healthcare Copilot"
_ACM.org/TheWebConf/2025/Conference — WWW 2025 Poster_

### Official Review · Reviewer_6BSR · 2024-11-28

**Novelty:** 6
**Technical Quality:** 6

**Review:**

The paper introduces MedRAG, a retrieval-augmented generation (RAG) framework designed for the medical domain. The framework enhances diagnostic reasoning and decision-making by integrating a four-tier hierarchical diagnostic knowledge graph (KG) with RAG models. MedRAG systematically retrieves relevant EHRs and utilizes KG-elicited reasoning within large language models (LLMs) to generate precise diagnoses, treatment recommendations, and follow-up questions. Evaluations on the DDXPlus public dataset and a private chronic pain dataset (CPDD) demonstrate MedRAG’s superior diagnostic accuracy and specificity compared to state-of-the-art RAG models.

strengths:
MedRAG’s use of a diagnostic KG addresses limitations in existing RAG frameworks by enabling fine-grained reasoning about diseases with similar manifestations. The hierarchical organization of the KG enhances its applicability to complex diagnostic scenarios. The inclusion of a proactive diagnostic questioning mechanism is innovative, allowing for the clarification of ambiguous or incomplete patient data. This feature demonstrates tangible improvements in diagnostic accuracy and consultation efficiency. The experiments cover both public (DDXPlus) and private datasets, illustrating MedRAG’s adaptability across different diagnostic contexts. The framework is also tested on a range of LLMs, showcasing its generalizability and compatibility. MedRAG consistently outperforms baseline models across multiple metrics, particularly in diagnostic specificity (L3 level). This demonstrates the effectiveness of integrating KG reasoning into the RAG pipeline.

**Questions:**

1. While proactive questioning improves accuracy, the paper does not elaborate on how well the generated questions align with clinical reasoning standards. Could the authors provide qualitative examples or a clinician’s evaluation of the questions generated?

2. The ablation studies indicate that KG-elicited reasoning significantly impacts performance. However, more detailed analysis of specific KG components (e.g., the diagnostic differences KG) could help identify their relative contributions to MedRAG’s overall effectiveness.

3. Is it fair to compare the QWEN with LLAMA in English corpus?

**Reviewer Confidence:**

3: The reviewer is confident but not certain that the evaluation is correct

**Scope:**

3: The work is somewhat relevant to the Web and to the track, and is of narrow interest to a sub-community

---

### Official Review · Reviewer_K92D · 2024-12-01

**Novelty:** 4
**Technical Quality:** 4

**Review:**

**Summary**

The paper presents the MedRAG framework with a four-layer hierarchical structure, which combines RAG and a comprehensive diagnostic knowledge graph to achieve more accurate reasoning and customized treatment recommendations. This approach significantly enhances RAG's reasoning capabilities, enabling it not only to identify subtle diagnostic differences but also to actively infer relevant follow-up questions.

**Pros**

1. The paper includes comprehensive experiments with both open and closed language models.
2. The paper has clear diagrams that accurately describe each part of this complex framework.
3. The mechanism of hierarchical clustering of diseases and the model actively asking questions is interesting.

**Cons**

1. How many errors are there in the process of Diagnostic Knowledge Graph Construction, and will these errors affect the results? Could you elaborate on how to prompt LLMs to make classification.
2. The L1 performance of some models in table2 seems to have decreased, what is the reason for this?
3. Could you provide more details about how proactive diagnostic questioning mechanism be tested.
4. It seems that the main evaluation metrics are based on accuracy. Could you provide any results of semantic metrics for analysis?
5. There is a typo in the model of Table2: GP-3.5-turbo

**Questions:**

Please see above.

**Reviewer Confidence:**

3: The reviewer is confident but not certain that the evaluation is correct

**Scope:**

4: The work is relevant to the Web and to the track, and is of broad interest to the community

---

### Official Review · Reviewer_8RwY · 2024-12-04

**Novelty:** 4
**Technical Quality:** 4

**Review:**

In this work, the authors propose MedRAG, a RAG model enhanced by a knowledge graph for medical LLM tasks. The experiments are conducted on two EHR datasets, one public and one private. The research task is important. My major comments are as follows:

1. The experiment settings lack details. What is the prediction task in the experiments? Is it disease prediction? Why is only accuracy reported? What EHR data are used in this work—only diagnoses, or are other elements such as lab tests and medications included? Are these data longitudinal or single-time? How are the EHR data fed into the LLMs?
2. Why are more common real-world EHR datasets or diagnostic QA datasets not used in this work, such as MIMIC or MedQA?
3. Why is the knowledge graph built from the EHR dataset rather than using other well-defined medical ontologies, such as UMLS?
4. The symbols in Section 4.2 are heavily annotated and difficult to follow. What is the insight behind the differences in KGs? Why not build an individualized knowledge graph like in [1]?
5. The details of the Top-k relevant EHR retrieval are unclear. Are these records from the same individuals? During the testing phase, can the model still access embeddings created during the training phase?
6. The figures in the paper show many outputs, but the authors only evaluate diagnosis.

[1] Jiang, P., Xiao, C., Cross, A. R., & Sun, J. GraphCare: Enhancing Healthcare Predictions with Personalized Knowledge Graphs. In *The Twelfth International Conference on Learning Representations*.

**Questions:**

Please refer to my comments above.

**Reviewer Confidence:**

3: The reviewer is confident but not certain that the evaluation is correct

**Scope:**

3: The work is somewhat relevant to the Web and to the track, and is of narrow interest to a sub-community